# Daily struggle to take antiretrovirals: a qualitative study in Papuans living with HIV and their healthcare providers

Elfride Irawati Sianturi [1,2] Elmiawati Latifah,[3] Ari Probandari,[4] Christantie Effendy,[5] Katja Taxis[1]

For numbered affiliations see end of article.

**Correspondence to**
Elfride Irawati Sianturi;
ira_sianturi@yahoo.co.id

## ABSTRACT

**Objective** The aim of the study was to explore the experiences of Papuans living with HIV to take antiretroviral therapy (ART) from the patient and the healthcare providers (HCPs) perspective.

**Design** This was a qualitative study covering all five tribes located in Papua Provinces one of two Indonesian provinces on Papua Island. Semistructured interviews were conducted with Papuans living with HIV and their HCPs. Interviews were transcribed verbatim and coded to find themes.

**Results** Overall, we conducted interviews with 13 Papuans living with HIV (mean age: 33 years, 61% female) and 14 HCPs (mean age: 42 years, 64% female) within five customary areas. HCPs included three physicians, nine nurses, two others. Two main themes were identified: (1) personal factors and (2) healthcare environment-related factors. Personal factors were beliefs and knowledge of ART, stigma from family, community and HCPs as well as practical problems such as transportation because of long distance. Within the theme of healthcare environment, the competences and attitudes of HCPs were particularly relevant. The findings are important in refining HIV treatment strategies implemented in Papua, especially when extending HIV care provided by community centres.

**Conclusions** Despite free access to ART, Papuans living with HIV struggle to remain on treatment. Considering local culture and religion in strategies to reduce stigma should be a priority.

## BACKGROUND

Antiretroviral therapy (ART) has saved millions of people living with HIV (PLHIV).[1] Treatment is lifelong. Despite such successes, access to treatment and retention in care is still an issue in many parts of the world.[2] Economic barriers, stigma, social relationships, religion, local concepts about illness and medication impact on the care for PLHIV.[3 4]

Indonesia consists of more than 17 000 islands and has been viewed as one of Southeast Asia's highest performing economies in recent decades.[5] A large number of islands have fostered the development of a diverse culture with more than four hundred ethnic

### Strengths and limitations of this study

► The findings provide new insights into the daily struggle in taking antiretroviral therapy among Melanesians in Indonesia.
► This study is the basis for quantitative studies to identify how widespread some of the issues are and the findings should inspire others to explore the local context which is important to know to develop strategies to strengthen primary healthcare in many parts of the world.
► This study specifically recruited patients from five different tribes in Papua. Other racial/ethnic groups may have different experiences.
► Recruiting patients through their healthcare professionals may create the risk that patients feel coerced to participate in the study.

groups, each with their own language(s).[5 6] The prevalence of HIV is about 0.1% across Indonesia. The prevalence in Papua is about 2.3%, with the highest prevalence found in the remote areas of the highlands.[7] Papua Island consists of two provinces, Papua and West Papua. Ethnically, most Papuans are Melanesians belonging to one of six distinct local tribes, with Christianity being the dominant religion. This is in contrast to the rest of Indonesia where Islam is the dominant religion. Within Indonesia, Papua has a special autonomy status due to high revenues from the exploitation of natural resources.[8 9] In many instances, Papuans are viewed as being primitive in a globalised word.[8] The feeling of being stigmatised seems to be common among Papuans and it has been growing due to inequalities and poor security level.[10] In particular, feeling stigmatised has been found in PLHIV since being infected means breaking cultural norms.[11–13] A prior study showed that stigma among female Papuans with HIV was even pronounced than male Papuans.[14]

Despite the status of a special autonomous region, challenges remain to organise

healthcare for Papuans living with HIV because of the large geographical area, the lack of infrastructure and transportation into many parts, especially the remote highlands. Other challenges include the high number of illiterates, especially in the remote areas[15] and the local concepts of illness, death and misfortune which include the widely held belief that death and sickness occur intentionally.[8 15] Such cultural concepts may result in communication barriers between healthcare workers who are mainly migrants from other areas of Indonesia and the native Papuans.[16] Healthcare for PLHIV was initially provided by hospitals, but in recent years, is also provided by community health centres.[16 17]

ART is provided free of charge in Papua.[18] To ensure the sustainability of the ART supply chain,[19–21] every PLHIV must be registered in the national cohort before initiation of ART.[22] Even though there were 153 locations which provided HIV care in Papua Province,[23] less than 25% of Papuans living with HIV were on ART.[7 23] The coverage of ART in Papua was lower than in other parts of Indonesia.[2] Previous survey-based studies in Papua showed low levels of health literacy,[24] and lower levels of adherence to medication than in other parts of Indonesia.[25] We also found that enacted stigma was significantly associated with non-adherence among Papuans.[26] To improve the situation for Papuans living with HIV, more in-depth knowledge is needed on barriers and facilitators to access medication and remain on treatment in particular in the changing organisation of healthcare taking into account the local culture. The aim of the study was to explore the experiences of Papuans living with HIV to take ART from the patient and the healthcare provider (HCP) perspective.

## METHODS
### Study design, setting and population
This was an exploratory qualitative study. Data were collected between July and September 2018. The study covered all five customary areas in Papua Province, namely LaPago (Wamena), Mee Pago (Timika), both from the highland areas and Anin Ha (Merauke), Saereri (Serui) and Mamta (Jayapura) from the lowland areas. We included patients who received ART either from a hospital or from a community health centre, and HCPs working in HIV care. Each site included at least two patients; one male and one female, and two HCPs. The inclusion criteria of patients were: (1) native Papuans or migrants having a Papuan spouse, (2) aged ≥18 years, (3) were on ART for at least 1 year before commencing the interview and (4) willing and able to give informed consent. The inclusion criteria of HCPs were: (1) working in HIV care for at least 2 years in a hospital or in a community health centre (called Puskesmas in Indonesia), (2) having completed training in HIV care and (3) willing and able to give informed consent. Before the interviews, participants were informed about the study purposes and ethical approval and signed consent was obtained from each participant. We assured all participants that data would be kept confidential and participants would have no disadvantages in receiving healthcare. This paper was written following the Consolidated Criteria for Reporting Qualitative Research guideline for reporting.[27]

### Study procedure, data collection and management of data
The interview guide was developed based on literature.[28 29] The research team of this study had a diverse background from pharmacy (EIS, EL and KT), epidemiologist (AP) and nursing field (CE) and all researchers are female. Two with a masters degree and three with a PhD degree. The interviews were conducted by EIS and two research assistants with a background in sociology and nursing who have worked for almost 25 years in HIV care in Jayapura. Interviews were conducted in Bahasa Indonesia which is the national language of Indonesia. All interviews took place in a private location as agreed between participants and interviewers, and lasted approximately 1 hour. Topics covered in the interviews were the health problems which triggered getting a test of their HIV status, and the patient's experiences before and after starting to take ART.

We audio recorded the interviews. The interviewers also took field notes during the interviews. All audio recordings were transcribed verbatim, leaving out noise sounds and conversation outside of the study topics. We limited transcription to three interviews per day per transcriber to ensure thoroughness. All transcripts were double checked. We offered the participants to read the transcripts, but none of the participations wished to do so. Each transcript was given a unique identifier composed of the type of participant and gender.

### Data analysis
Data were analysed using the Atlas Ti V.8.4 software. Transcripts were coded by EIS, EL and CE taking a similar approach as in a previous study. We chose a content-oriented approach to analyse the data (undirected coding). All transcripts were read and reread to develop an initial coding framework which was refined during coding. Once the coding was completed, categories were developed. Themes were identified from the categories. Disagreements in the analysis were resolved by discussions between all authors. The themes were finally mapped with an existing larger framework.[28 30] We stopped interviewing when we reached saturation within two participant groups.

### Patient and public involvement
EIS has worked as an HCP in Papua previously and therefore knew HPCs in all facilities. EIS contacted potentially eligible HCPs in all facilities and invited them to participate in the study. HCPs agreed to participate were also asked to contact potentially eligible patients to participate

**Table 1** Characteristics of study participants (n=27)

| Participant | Characteristics | No (%) | Mamta (Jayapura) | Mee Pago (Timika) | Saereri (Serui) | Anin Ha (Merauke) | LaPago (Wamena) |
|---|---|---|---|---|---|---|---|
| Healthcare providers (n=14) | Mean age 42 (range: 35–55 years) | | | | | | |
| | Work at hospital | 8 (57) | 1 | 1 | 3 | 2 | 1 |
| | Work at community health centre (*Puskesmas*) | 6 (43) | 2 | 2 | 1 | 0 | 1 |
| | Female | 9 (64) | 1 | 3 | 2 | 1 | 2 |
| | Male | 5 (36) | 2 | 0 | 2 | 1 | 0 |
| | Physician | 3 (12) | 1 | 0 | 1 | 1 | 0 |
| | Midwife | 1 (7) | 0 | 0 | 1 | 0 | 0 |
| | Nurse | 9 (64) | 2 | 3 | 1 | 1 | 2 |
| | Psychologist | 1 (7) | 0 | 0 | 1 | 0 | 0 |
| Patients (n=13) | Mean age 33 (range: 20–60 years) | | | | | | |
| | Medication dispensed from hospital pharmacy | 10 (77) | 3 | 0 | 3 | 3 | 1 |
| | Medication dispensed from community health centre (*Puskesmas*) | 3 (23) | 0 | 2 | 0 | 0 | 1 |
| | Female | 8 (61) | 2 | 1 | 2 | 2 | 1 |
| | Male | 5 (39) | 1 | 1 | 1 | 1 | 1 |
| | Diagnosis following VCT | 11 (85) | 3 | 2 | 2 | 3 | 1 |
| | Diagnosis on antenatal ward | 2 (15) | 0 | 0 | 1 | 0 | 1 |

Puskesmas, Pusat Kesehatan Masyarakat; VCT, voluntary, counselling and testing.

in this study. The participants gave written informed consent before interviews.

## RESULTS
### Participants
All HCPs and PLHIV who were approached agreed to participate in the study. Overall, 14 HCPs participated who recruited 13 patients (table 1). The 14 HCPs were three physicians, nine nurses, one psychologist and one midwife. The mean age of HCPs was 42 years. Among HCPs, nine were female and eight worked in hospital. Ten out of 13 Papuan patients got their medication from the hospital pharmacy, and the remaining patients accessed *Puskesmas* to collect their ART. The mean age of patients was 33 years. Among the patients, 8 were females and 11 patients were diagnosed in a voluntary, counselling and testing clinic following ongoing symptoms of illness. There was no participant who was contacted refused to participate this study. Additional participant's characteristic shows in table 1.

The codes and categories were mapped on the two themes from existing frameworks.[28 30] The themes were: (1) factors that represented patient's personal experiences of taking ART and (2) the descriptions of the healthcare environment which played a role in taking ART.

### Theme 1: patient factors
We identified personal beliefs and knowledge about ART, religion, verbal and nonverbal communication, finances and transport, social support and other responsibilities as being important factors for patients in dealing with taking ART.

### Personal beliefs and knowledge about ART
A patient who previously dropped out of care described that they believed in traditional medicine being helpful. The patient also believed ART was a not a drug since ART did not cure the symptoms and the disease.

> My doctor said that my ART required a lifetime commitment. Then I thought ART was just to extend my

life not for making my virus disappear…[Patient, Male]

One HCP explained insufficient knowledge has challenged patients to try alternative medication.

The insufficient knowledge is a problem for the patients. I found patients attempting to use traditional medication since they had the lack of knowledge of ART. I think it was because we informed them that their ART is to suppress their virus not to eliminate. [HCP, Female]

Differently, the patient's knowledge level played a role in coping with HIV.

### Religion

Most participants took religion seriously. Christian belief shaped attitudes of participants to accept HIV as their disease. Another patient took hope from religious belief that HIV could be cured. They believed that taking ART was the way for God to help them.

God's miracle will happen, just believe and take ART regularly. [Patient, Male]

In a similar way, HCPs used religion to measure the quality of their service and believed helping patients was the same as serving God.

I told my colleagues, our salary and incentives would be bloody money, when we were not honest in doing our job. Our money would be like water into our no-ken (woven bag). It disappears fast because we did not work according to God's will. We are blessed if we treat our patients likely what God wants. [HCP, Female]

In contrast, both patients and HCPs also shared their views that based on the religion patients were immoral, and God used HIV as a warning sign to sinners.

I think there is a link between HIV and religion. Patients were infected with the virus because they were not afraid to of God's law. They were not married yet, drunk and had sex with women who were not their partner. [HCP, Female]

### Verbal and nonverbal communication

Participants stressed it was important to talk about genital terms in prudent ways as this is important for Papuans. Besides verbal language, patients reported about nonverbal signs such as unprofessional behaviour of personnel and breaches of confidentiality. One patient strongly articulated his experiences with an HCP. Laughing and not being seriously while informing about disease related to sexual transmitted disease was found and irritated PLHIV.

A nurse increased her voice when giving us information…it might be her character…. However, I preferred that others who did not have business with my health could not hear [Patient, Male]

The language HCPs used was mentioned as an issue. For example, HCPs had problems to express medical terms about side effects in simple words, especially HCPs who were not native Papuans.

I think language is challenging here. I tried counselling them about side effects, such as rash, or stiffness but I could not find words to replace those into their language. So, sometimes I must say 'kaskado' to replace rash. [HCP, Female]

The gap in language has also produced misunderstandings in the community, especially in the HIV awareness programme. One patient shared his experience about this programme before he was diagnosed with HIV as follows.

The nurses came here often to give some programs about the HIV awareness. They said many things, we listened but we got nothing. We did not understand ….[Patient, Male]

### Finance and transport

Patients weighed the costs and benefits of being treated in *Puskesmas*, however, some patients preferred the hospital despite having to spend more money and time.

I needed about IDR [Indonesian rupiah] 30.000–60.000 every time to collect my ART at a hospital. It may be costlier than collecting ART at *Puskesmas*, which only take 10.000 IDR, but I decided to keep collecting ART here in hospital. [Patient, Female]

In general, participants said that distance and transportation cost were barriers to keep taking ART.

I had been waiting for the trucks which would bring me to the city to collect my ART. Nevertheless, my boss and the trucks never came, and I heard that there was a problem with gasoline stock supply. [Patient, Male]

### Social support from family and peers

We found that patients and HCPs had different opinions about social support. In general, a positive relationship with others was described as powerful to support patients over time and reduce the risk of discontinuation of ART. There was both, acceptance and rejection from families. Even though it happened, HCPs considered having family support was essential to be present before they placed patients on ART. The HCPs believed the existence of family would help patients to overcome fear about medication and social problems.

Most of the patients who had a good relationship with family, they are adherent and there is only a small percentage of them who stopped their ART. However, for patients who hide their status, most of them found their lives as useless and they could stop ART anytime. The hindrances do not really impact patients when they know their family protect them. [HCP, Female]

Patients revealed different feelings and experiences about the family's support. They expressed that the family influenced the decision to seek healthcare. There could be a delay in being diagnosed and be on ART because of the family. One patient mentioned her family brought her to hospital, which was far away from her house. It was impossible to bring her to the nearest health facility because her father also worked there.

In order to receive help from their family, patients needed to disclose their status. To disclose HIV status was described as a double-edged sword, either to receive support or being stigmatised.

> My family know my HIV, they support me to take my medicine, but they have separated my plate, glasses from others. [Patient, Male]

Participants also reported about their experiences with support from peers, that is, other PLHIV. In some locations formal or informal peer-support groups exist. Some patients described that knowing peers helped them to cope with fear and empower them. With peers, patients were not alone anymore and sometimes peers linked them with better HIV care.

> I join peers. I am happy with that. I can meet people who have the same condition with me. We discussed many things, not only medication but how we deal with stress. With them I can discuss many things including something private, ha ha…about our genitals. [Patient, Female]

One patient revealed that an HCP asked her to visit patients who did not attend the hospital for collecting ART.

> In my situation, HCP asked me to visit other patients who did not attend the hospital. HCP considered my visit to patient's house could not raise suspicions. All my expenses to visit other patients were paid by HCP. [Patient, Female]

HCPs said that the influence of peers may be positive or negative.

> Sometimes, I was disappointed with some peers. They persuaded my patients who were adherent to move to other health facilities and asked them to try herbals. I know it was not 100% peer's faults, my patients could not filter obtained information but the reason we introduced peers to patients was to help others but to not make others their followers. [HCP, Female]

### Other responsibilities

A number of difficulties keeping to the regular medication intake schedule were identified. These included being bored or having other responsibilities, for example, work-related activities which made it difficult to take ART. One patient said,

> I don't think so that my ART is not important, but I must sell 'pinang' (betel nut) every day to support

my life and my daughter. I worked until late at night and it made me felt tired and sleepy. Then I forgot to take my medication. So if I have a break I take my medication directly. [Patient, Female]

### Theme 2: healthcare environment

Within the theme of healthcare environment, we identified health service and HCPs as important factors which influenced the experiences of patients in taking ART. Infrastructure and perceived complicated administrative system were most commonly reported as health service-related factors. Participants also shared their experiences with the home visit programme and educational services. Provider factors consisted of willingness to help, competence of HCPs and trust in HCP.

### Health service factors
#### Infrastructure

Both patients and HCPs reported *Puskesmas* and hospital had the facilities to diagnose HIV. Some patients did not know that it was possible to collect ART in *Puskesmas*. Even though there is the possibility of collecting ART in *Puskesmas*, this option was rejected by some patients. Patients found the infrastructure and laboratory monitoring insufficient. A community centre with only a small room used for all patients irrespective of their disease created fear of being exposed as HIV positive to others.

> I think the performance of personnel in Puskemas is good. However, the consultation room is small and without separation. Everyone can hear what nurses are saying… I think everyone can recognize my HIV directly. [Patient, Male]

The underutilisation of *Puskesmas* has been considered as problems for patients and HCPs. The HCPs were frustrated because they realised that their goal of improving the quality of health services could not be achieved.

> Since I have worked in *Puskesmas*, I have found the lack of laboratory is still a problem. We want that *Puskesmas* can be independent and our goal to deliver the high quality of health service can be achieved. I do not know when it is going to happen. [HCP, Female]

#### Perceived complicated administration system

Patients and HCPs mentioned administration as an important topic. Most patients could not understand why HCPs asked them to show their single identity number (*Nomor Induk Kependudukan*, NIK). The patients thought that this was irrelevant since costs for healthcare should be covered. One patient expressed her worries about this:

> My concern was only about my card…… My nurses have asked it many times and I could not show it. [Patient, Female]

Some HCPs explained that patients needed to register with their NIK to be included in the national HIV cohort

to receive ART. Without NIK, a delay in starting ART could occur.

### Home visit programme and educational services

Patients and HCPs also discussed the effectiveness of the home visit programmes. HCP's perspective revealed that they needed to attach some pictures as proof to get reimbursement from their health insurance. However, patients felt uncomfortable about the HCPs taking pictures in their homes. Also, patients felt the home visits increased the suspicion in the community. Some patients offered to have meetings with HCPs in another place but not in their house.

> I remembered one of my patients refused my visit to his house. He said to me…. 'please do not come…. I am afraid people will be curious about your visit'. He offered to meet me in other place instead of his house. [HCP, Female]

Sometimes HCPs found their colleagues documented their activities to make fictitious visits. One HCP shared,

> Most of our colleagues tried making fictitious programs. They made some photos as complementary documents to get reimbursement. Most of our colleagues focused on money and they had many tricks to get reimbursement. [HCP, Female]

Some HCPs noted that they often had a deterrent effect when promoting the HIV programme in community because they used scary images of skeletons. The concept of HIV could be treated with medication was omitted from the programme.

> In early days, we used images of death, skeleton and disfigurement in our HIV awareness program. We hope the community can get a deterrent effect. Later on, the community can learn about how to prevent it. However, we found the information about the HIV syndrome was among the topics most commonly remembered by community. I think most people are stubborn. It is better to use a deterrent effect. We could not say our information about HIV in sweet words. It could not work. [HCP, Female]

> When we have time to inform about ART that we could not say more such as ART could suppress HIV and could not cure of HIV. Of course, we would like to inform patients and community completely. However, we should be wise to consider our targeted individuals. We say ART is a drug because if we send all information, patients and community could remember only the small part of the end our information. [HCP, Male]

### Provider factors
#### Willingness to help

In general, the willingness of HCPs to help patients was strongly articulated from both patients and HCPs. For example, HCPs took into account personal circumstances to help patients. One HCP said,

> If their home is far from the hospital, particularly for patients who fly by plane, we can give them 2 or 3 months of their ART. I appreciated a pregnant woman who travelled by bus from Sarmi [the name of a district] to here. I can see her commitment to prevent her baby from having HIV. [HCP, Female]

On the other hand, HCPs had to consider the consequences of their help such as providing more supply of ART to one patient. One HCP said,

> Then after giving the stock for three months, we should be cautious about the stock for other patients. Giving three months ART for patients means we borrow from other patient's stock. [HCP, Female]

Another HCP provided other help such as creating a safe environment in the clinic, transportation cost and providing food, as one explained.

> Here, I feel safe, everybody knows each other and I don't need to hassle for collecting ART, I come here every two months. [Patient, Male]

However, patients also acknowledged the fear of rejection and being stigmatised. One patient explained:

> How they treated me so bad, I must fight in emergency department after they rejected me. I was there not for free. I am Papuan, and I have rights to get health access. Finally, one doctor came then I said to him that I did not want others in that hospital to discriminate more. It was enough. [Patient, Female]

### Competence of HCPs and trust in HCP

Patients recognised the competence of HCPs, for example, to provide information about medications. Patients described the benefits of ART. A patient described why she needed ART for the rest of her life.

> If someone is asking me about ART, I can let them know that ART can suppress my virus but not to cure. I need to take this tablet for a lifetime. My nurses explained this often to me. [Patient, Female]

Similarly, HCPs perceived they had power to make patients feel inferior and dependent on healthcare system without any possibility to protest.

> I felt hurt, when I saw my colleagues treating patients while laughing and sometimes they were busy with their mobile during consultation. [HCP, Female]

One patient revealed her situation when she wanted to receive information from her health providers.

> I kept my mouth suddenly, when I saw her face. I could not open my mouth even I really needed to ask her about my problem. [Patient, Female]

One HCP revealed problems with confidential patient information. It was commonly found that HCPs refused to treat patients and asked other colleagues to replace their shift in preparing medication.

A few weeks ago, we had problems with our staff. One of our staff was unable to maintain patient's status. He told the status of the patient to the family although the patient asked us to keep it confidential. He was not ready yet to disclose the status. I can understand if he disappeared and did not collect medication for a few months. But everything is now solved. [HCP, Female]

## DISCUSSION

In this qualitative study in Papua province, a region with a high prevalence of HIV and a relatively low uptake of ART, we explored Papuan's experiences of taking ART from the patient and the HCP perspective. We found that patient's knowledge, beliefs and routines and the healthcare environment had an impact on the medication taking of patients.

In line with a previous study, we found that just the availability of ART is insufficient for patients to cope with their chronic HIV treatment.[31] In recent years, healthcare has been reorganised, so that HIV care can be provided in remote areas by *Puskesmas*, but some patients do not use those because of a lack of privacy. As in other studies, fear of disclosure of HIV status and the stigma associated with disclosure was an important barrier to accept this care closer to home.[32] This was not only found to be a barrier to get supply with ART, but also to discuss concerns and get information on the disease and treatment.[31 33]

Contrary to a previous study,[34] patients valued the competence of HCPs highly. Nonetheless, both HCPs and patients reported communication barriers. An example was the difficulties to describe side effects of ART with sufficient detail, since the local language lacked the terms to do so. In addition, patients experienced the attitude of some HCPs as degrading. Examples included HCPs laughing or not paying attention to patients or discussing sensitive issues like genitals directly and loudly, as have been shown in other studies.[8 35] Similar as in other studies, use of educational materials producing fear was also perceived to contribute to stigma.[9 36] Since having training before HCPs placed is mandatory, this study recommends stigma reduction should be available in one of training topics. Addressing these issues in training of HCPs and developing educational materials suitable for the local population seems to be a first step to tackle these problems.

Patients and HCPs agreed that support from family, HCP and peers were important for the patients to cope with treatment. In particular, the support of the family in medication taking was seen as essential as in other studies.[37–39] However, despite support, patients could be feeling isolated by their family.[40] This condition may

lead them to keep their HIV status as a secret. As in other studies, peer support was found to help patients.[41] But peers should receive education and training, because there is a risk of misinforming patients.[42] Furthermore, HCPs experienced difficulties to find sufficient patients who wanted to support their peers because of fear to disclose their status. HCPs recognised that support should be personalized for each patient.

Religion was an important topic to be discussed for patients and HCPs. Contrary to a previous study,[42] the religious beliefs empowered patients to take ART and being hopeful. Religious beliefs were also an important motivator for HCPs to offer the best service to patients. However, religious beliefs also contributed to the notion that HIV was a punishment due to personal failure.

This study highlighted that feeling stigmatised was widespread among PLHIV as stigma seemed to be the underlying problem for many of the issues that we identified.[43] A stigma reduction programme seems to be needed to maximise the effects of existing HIV policies and provision of ART. There is very limited information how to successfully reduce HIV stigma in Indonesia. However, some interventions among people affected with leprosy in Indonesia[44 45] might be good examples to be implemented. Testimony, and counselling, significantly reduced internal and external stigma among leprosy patients and the community. Therefore, testimonies of HIV infected women took ART and were successful in having babies with no HIV infection might change the perceptions of community and HCPs. Furthermore, strengthening the economic situation of leprosy patients by providing microfinance has been shown to be successful. Similar with previous studies,[46 47] people who were in a better socioeconomic situation had better self-esteem and were actively involved with their community and had less internal stigma.

### Strengths and limitations

It should be noted that there were some limitations in this study. First, we specifically recruited patients from five different tribes in Papua. Other racial/ethnic groups and may have different experiences. Since we only included few participants from each tribe, we were not able to ascertain similarities and differences between tribes. Our data show the overall experiences of PLHIV in this region. Second, we asked HCPs to ask patients to participate in this study, so our results refer to patients who remain in chronic care. More work needs to be done to identify factors driving patients to drop out of care completely. Third, recruiting patients through their healthcare professionals creates the risk that patients feel coerced to participate in the study. Because of the topic and non-obtrusive nature of this study, we believe this risk was minimal, but we cannot exclude this completely. Despite these limitations, our findings provide new insights into the daily struggle in taking ART among Melanesians in Indonesia. Our findings are the basis for quantitative studies to identify how widespread some of the issues are. Furthermore, our findings should inspire

others to explore the local context which is important to know to develop strategies to strengthen primary health-care in many parts of the world.[48]

## CONCLUSION

Our findings are important for the further development of interventions to support patients in chronic treatment with ART taking into account the specific cultural needs of Papuan with HIV. In particular, addressing communication barriers is important. The HIV treatment strategies implemented in Papua need to take into account the local social and religious culture of Papuans. Broader strategies to reduce stigma should be a priority for the centralised and local government. This may remove some of the barriers for PLHIV to use the primary health centres. Hopefully, this can improve retention and adherence, and ultimately, health outcomes for Papuans living with HIV.

**Author affiliations**
[1]Groningen Research Institute of Pharmacy Unit PharmacoTherapy, -Epidemiology & -Economics (PTEE), Groningen University, Faculty of Mathematics and Natural Sciences, Groningen, The Netherlands
[2]Department of Pharmacy,Faculty of Mathematics and Natural Sciences, University Cenderawasih, Jayapura, Indonesia
[3]Department of Pharmacy, Faculty of Health Science, Universitas Muhammadiyah Magelang, Magelang, Indonesia
[4]Department of Public Health, Faculty of Medicine, Universitas Sebelas Maret, Surakarta, Jawa Tengah, Indonesia
[5]School of Nursing, Faculty of Medicine, Universitas Gadjah Mada, Yogyakarta, Indonesia

**Acknowledgements** The authors would like to express their gratitude to Papuans living with HIV and healthcare providers who participated in this study.

**Contributors** Conceived and designed the study: EIS, KT, EL, CE and AP. Analysed the data: EIS, KT, EL, CE and AP. Wrote the paper: EIS, KT, EL, CE and AP. All authors read and approved the final manuscript.

**Funding** This work was supported by Directorate General of Higher Education (DIKTI) Scholarship provided to EIS (No:15.6/E.4.4/2015).

**Disclaimer** The funding sponsor had no role in the design of study; in the collection, analyses, or interpretation of the results; in writing section, and the decision to publish this.

**Competing interests** None declared.

**Patient consent for publication** Not required.

**Ethics approval** The study was approved by the Ethics Commission, Faculty of Medicine, Public Health, and Nursing Universitas Gadjah Mada (number: KE/FK/0507/EC/2018).

**Provenance and peer review** Not commissioned; externally peer reviewed.

**Data availability statement** Data are available on reasonable request. All data relevant to the study are included in the article or uploaded as online supplemental information. The data would not be shared outside of participating research institutions. Any queries on how to access the data set should be to the corresponding author or ira_sianturi@yahoo.co.id.

**ORCID iD**
Elfride Irawati Sianturi http://orcid.org/0000-0001-8562-9956

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
