## [Reviewer comments · BMJ Open]

ARTICLE DETAILS

TITLE (PROVISIONAL)	The daily struggle to take antiretrovirals: a qualitative study in Papuans living with HIV and their healthcare providers
AUTHORS	Sianturi, Elfride; Latifah, E; Probandari, Ari; Effendy, C; Taxis, Katja

VERSION 1 – REVIEW

REVIEWER	Jennifer Watermeyer University of the Witwatersrand, South Africa
REVIEW RETURNED	28-Jan-2020

GENERAL COMMENTS	Thank you for the opportunity to review this paper. It reports on a qualitative study that explored patient and HCP experiences of HIV care in a particular Indonesian context. While some of the results are interesting and the quotes make for compelling reading, unfortunately I do not believe the article is ready for publication. My primary concern relates to the way in which the data has been analysed and interpreted, and I elaborate on this below. The paper needs considerable editing. In addition, the original/novel aspects of the study need to be made apparent – although it might be of interest to interview participants in a particular context, the rationale for choosing this context needs to be enhanced and potential implications of the findings for other contexts explored. Further, there are a number of ethical concerns with the study, and there is no mention of ethical approval via an IRB and how consent was obtained from participants. There is no theoretical framework nor any relevant concepts referred to in this paper. It would be useful if the authors could be more specific about what they mean by 'experience' – lived experiences of living with HIV, or experiences of care? The research question is not stated and the aim/objective needs to be more specifically worked out. The literature review needs expansion, with particular focus on providing an overview of previous literature from other contexts that speaks to experiences (of care) of people living with and working with HIV. The methods lack sufficient detail. It is not clear why participants were sampled from across 5 tribes, but then so few participants were included from each tribe. It is not clear where participants were recruited from – clinics or hospitals? There is no mention of ethical considerations or the recruitment process. It would appear that some participants may have been coerced into participating because they knew the researchers – what measures were put in
---

	place to counteract coercion? The topics covered in the interviews do not seem to relate strongly to perceptions and experiences in general, but quite specifically to experiences of care. In what language were the interviews conducted? The analysis process is vague. There is no mention of a particular analytic method (for example, Braun and Clarke's thematic analysis). Atlas Ti is a tool for analysis, not an analytic method. The analysis needs revisiting. Many of the quotes included in the table in Appendix 2 do not relate to the overarching theme, and the description of each theme does not always make sense. For example, much of Theme 1 seems to relate to stigma and not necessarily 'organisation of care'. The illustrative quotes in Appendix 2 for Theme 1 seem to relate to a lack of resources, not necessarily organization of care (do the authors mean 'organizational routines' here?). Theme 2 seems to relate more to communication and relationship issues between healthcare providers and patients. Theme 3 again refers to stigma issues. The themes are too broad and vague. There is overlap between the themes (e.g. Themes 2 and 3). Subheadings are used only sometimes in the Results section, but in general there are too many sub-themes/categories described within each broad theme and some of these do not seem to relate to the main theme. What is required is a more nuanced approach to analysis and interpretation of the data, and I would suggest the authors have a good look at Braun and Clarke's work on analysis and at some of their examples of how to do analysis. It is not clear what is meant by a 'peer'. Do the authors mean something similar to the accompagnateur model described in Haitian HIV care, for example? The Discussion section requires a more nuanced approach rather than merely repeating the results and relating to literature. I wish the authors well in revising and reworking this paper.
--	--

REVIEWER	Cynthia D. Fair Elon University
REVIEW RETURNED	08-Feb-2020

GENERAL COMMENTS	The daily struggle to take antiretrovirals: a qualitative study in Papuans living with HIV and their healthcare providers Thank you for the opportunity to review your manuscript on this vital subject. Understanding the factors that are associated ART adherence is important. Further this is understudied population deserving of attention. Abstract: There is a word missing in the objective: Objective: The study aimed to explore the perspective and the experiences of Papuans, are Melanesians with Christianity as the dominant religion, living with HIV to take ART. Background:
---

	The introduction provides an overview of HIV in Indonesia and the cultural context of Papuans. Please cite the claim that “The feeling of being stigmatized seems to be common among Papuans.” Is that in general, or primarily related to HIV? Can you further explain this statement: “We also found that stigma was significantly associated with non-adherence [23].” What kind of stigma (internalized?) and what kind of non-adherence? Please clarify the objective: “This study aimed to explore the perspective and the experiences of Papuans living with HIV and their strategies to take ART.” Do you mean strategies used to maintain adherence to ART? Methods Can you explain how the qualitative codes were developed? More detail here would be helpful. Results Consider changing the first sentence to: “All HCPs and PLHIV who were approached agreed to participate in the study.” Consider adding more information to your introduction of the 4 themes. - Four themes related to ART adherence emerged including, -It seems as though there are different levels of influence. Most distal to the person LIHV is the organization of care and closest to the person would be their knowledge, beliefs, etc. Can you comment on the nature of the factors which would be a deeper analysis of the data. Each theme needs to be contextualized. As it stands the authors present a theme and then primarily offer direct quotes to support that theme. This approach makes the results feel choppy and less coherent. The paper would be stronger with more reflection about the theme itself followed by supporting evidence. Please clarify this statement: “It was commonly found that HCPs refused to treat patients and asked other colleagues to replace their shift in preparing medication.” How common? This is quite a shocking result and should be further explored and addressed in the discussion. I was confused by this statement: “However, HCPs added the existence of peers was diverse and misinterpretation of patient to response about peer was commonly occurred.” Are you saying peer-to-peer support was diverse? Please explain what you mean by the entire statement. Your statement, “Patients who had sufficient knowledge revealed that being healthy and being adherent reduced stigma from others”
--	--

	implies a threshold of knowledge. Did you assess level of knowledge? What does that mean in the context of your study? Same with “insufficient” level. You use the word “Interestingly” to begin many sentences in the results section. Please use it judiciously. Discussion Please explain what you mean that your findings are “more diverse than a previous study”. I’m confused by the word “diverse”. Your discussion would be stronger if you made clear and direct recommendations based on your findings to address the different levels of factors that influence ART adherence. What organization changes need to be made? How can health care providers improve? How can PLHIV experience increased social support? What strategies can improve the ART knowledge of PLHIV? This is an important study. The policy and clinical implications should directly reflect your findings. Best of luck with your work.
--	--

VERSION 1 – AUTHOR RESPONSE

AUTHOR RESPONSE TO REVIEWER 1

Reviewer Name: Jennifer Watermeyer

Institution and Country: University of the Witwatersrand, South Africa

Please state any competing interests or state ‘None declared’: None declared

Please leave your comments for the authors below

Thank you for the opportunity to review this paper. It reports on a qualitative study that explored patient and HCP experiences of HIV care in a particular Indonesian context.

-While some of the results are interesting and the quotes make for compelling reading, unfortunately I do not believe the article is ready for publication. My primary concern relates to the way in which the data has been analysed and interpreted, and I elaborate on this below. The paper needs considerable editing.

Response: We thank the reviewer for the useful comments and we give a detailed response to the individual comments below. Furthermore, we have carefully edited the paper.

In addition, the original/novel aspects of the study need to be made apparent – although it might be of interest to interview participants in a particular context, the rationale for choosing this context needs to be enhanced and potential implications of the findings for other contexts explored.

Response: We have amended the introduction, methods and discussion with more details to explain the rationale for the study and the potential implications of the study findings.

Further, there are a number of ethical concerns with the study, and there is no mention of ethical approval via an IRB and how consent was obtained from participants.

Response: We have obtained approval for the study by the Ethics Commission, Faculty of Medicine, Public Health, and Nursing Universitas Gadjah Mada. (Page 5 Line 24).

We have obtained consent by informing them about the study purposes and ethical approval and signed consent was obtained from each participant. (Page 5 Line 26)

There is no theoretical framework nor any relevant concepts referred to in this paper.

Response: Thank you. We have used a theoretical framework to map our findings. We have amended the methods section to explain this in more detail. (Page 6 Line 24)

-It would be useful if the authors could be more specific about what they mean by 'experience' – lived experiences of living with HIV, or experiences of care?

Response: The focus of our study is on the experiences in taking antiretroviral treatment. We have therefore rephrased study aim to make this explicit. (Page 5 Line 9)

The research question is not stated and the aim/objective needs to be more specifically worked out.

Response: We have rephrased the aim of the study to be more specific what we have studied. (Page 5 line 9)

The literature review needs expansion, with particular focus on providing an overview of previous literature from other contexts that speaks to experiences (of care) of people living with and working with HIV.

Response: We have added a number of references which address the experiences of PLHIV in taking their medicines. However, we would prefer to keep the introduction focused and not too long.

-The methods lack sufficient detail. It is not clear why participants were sampled from across 5 tribes, but then so few participants were included from each tribe. It is not clear where participants were recruited from – clinics or hospitals?

Response: We have amended the methods section to include more details on the recruitment process. We have sampled participants from across different tribes to get a diversity of experiences in taking ART. Since we only included few participants from each tribe, we were not able to explore differences and similarities between the tribes. Our study may be a starting point to explore this in more detail. We have amended the limitation section accordingly. (Page 16 line 24)

There is no mention of ethical considerations or the recruitment process. It would appear that some participants may have been coerced into participating because they knew the researchers. What measures were put in place to counteract coercion? The topics covered in the interviews do not seem to relate strongly to perceptions and experiences in general, but quite specifically to experiences of care.

Response: We have amended the methods section with more details on the recruitment process. We have also amended the limitation section to discuss concerns that participants may have been coerced into participating in the study.

We wrote: "Third, recruiting patients through their health care professionals creates the risk that patients feel coerced to participate in the study. Because of the topic and non-obtrusive nature of the study we believe this risk was minimal, but we cannot exclude this completely." (Page 16 line 31)

In what language were the interviews conducted?

Response: Additional information has been added. In this study, all interviews were conducted in Bahasa Indonesia and it has been written in method section on page 6.

We wrote: "Interviews were conducted in Bahasa Indonesia as a national language." (Page 6 Line 4)

The analysis process is vague. There is no mention of a particular analytic method (for example, Braun and Clarke's thematic analysis). Atlas Ti is a tool for analysis, not an analytic method.

Response: Thank you. We have amended the methods section to provide much more details on the analysis process.

-The analysis needs revisiting. Many of the quotes included in the table in Appendix 2 do not relate to the overarching theme, and the description of each theme does not always make sense. For example, much of Theme 1 seems to relate to stigma and not necessarily 'organisation of care'. The illustrative quotes in Appendix 2 for Theme 1 seem to relate to a lack of resources, not necessarily organization of care (do the authors mean 'organizational routines' here?). Theme 2 seems to relate more to communication and relationship issues between healthcare providers and patients. Theme 3 again refers to stigma issues.

Response: We agree with the reviewer that some of the information provided in Appendix 2 was confusing. We have therefore decided to remove Appendix 2. We have amended the methods section as highlighted above to give more details of the analysis which we hope is helpful in understanding our manuscript. We would like to emphasize that stigma is an important code which played a role across all the different themes.

-The themes are too broad and vague. There is overlap between the themes (e.g. Themes 2 and 3). Subheadings are used only sometimes in the Results section, but in general there are too many sub-themes/categories described within each broad theme and some of these do not seem to relate to the main theme. What is required is a more nuanced approach to analysis and interpretation of the data, and I would suggest the authors have a good look at Braun and Clarke's work on analysis and at some of their examples of how to do analysis.

Response: Thank you. We have re-analyzed the data and we have restructured the results section (also based on comments by reviewer 2).

-It is not clear what is meant by a 'peer'. Do the authors mean something similar to the accompagnateur model described in Haitian HIV care, for example?

Response: we have amended the results and discussion section to explain what we mean by peer.

-The Discussion section requires a more nuanced approach rather than merely repeating the results and relating to literature. I wish the authors well in revising and reworking this paper

Response: We have amended the discussion section to discuss better the implications of the findings.

AUTHOR RESPONSE TO REVIEWER 2

Reviewer Name: Cynthia D. Fair

Institution and Country: Elon University

Please state any competing interests or state 'None declared': None declared

Please leave your comments for the authors below

The daily struggle to take antiretrovirals: a qualitative study in Papuans living with HIV and their healthcare providers

Thank you for the opportunity to review your manuscript on this vital subject. Understanding the factors that are associated ART adherence is important. Further this is understudied population deserving of attention.

Abstract:

There is a word missing in the objective:

Objective: The study aimed to explore the perspective and the experiences of Papuans, are Melanesians with Christianity as the dominant religion, living with HIV to take ART.

Response: We have rephrased the aim of the study in response to comments by reviewer 1. (Page 5

line 9)

Background:

The introduction provides an overview of HIV in Indonesia and the cultural context of Papuans.

-Please cite the claim that "The feeling of being stigmatized seems to be common among Papuans." Is that in general or primarily related to HIV?

Response: we have edited this sentence to provide clarity and added a reference. (Page 4 Line 17)

-Can you further explain this statement:

"We also found that stigma was significantly associated with non-adherence [23]." What kind of stigma (internalized?) and what kind of non-adherence?

Response: Thank you. We have rephrased this sentence to provide more clarity. (Page 5 Line 5)

-Please clarify the objective:

"This study aimed to explore the perspective and the experiences of Papuans living with HIV and their strategies to take ART." Do you mean strategies used to maintain adherence to ART?

Response: We have rephrased the aim of the study also in response to comments by reviewer 1. (Page 5 line 9)

-Methods

Can you explain how the qualitative codes were developed? More detail here would be helpful.

Response: We have amended the methods section to describe better our analysis process also in response to comments by reviewer 1.

-Results

Consider changing the first sentence to:

"All HCPs and PLHIV who were approached agreed to participate in the study."

Response: we followed your suggestion.

We wrote: "All HCPs and PLHIV who were approached agreed to participate in the study" (Page 7 Line 3)

Consider adding more information to your introduction of the 4 themes.

- Four themes related to ART adherence emerged including,.....

-It seems as though there are different levels of influence. Most distal to the person LIHV is the organization of care and closest to the person would be their knowledge, beliefs, etc. Can you comment on the nature of the factors which would be a deeper analysis of the data.

Response: We have amended the methods section to describe better our analysis process.

-Each theme needs to be contextualized. As it stands the authors present a theme and then primarily offer direct quotes to support that theme. This approach makes the results feel choppy and less coherent. The paper would be stronger with more reflection about the theme itself followed by supporting evidence.

Response: Based on this comment and suggestions by reviewer 1, we have re-analysed our data and have restructured the result section following an existing framework by Holtzman and Anderson. We believe that our results are easier to follow and understand now.

-Please clarify this statement:

"It was commonly found that HCPs refused to treat patients and asked other colleagues to replace their shift in preparing medication." How common? This is quite a shocking result and should be further explored and addressed in the discussion.

Response: We agree that this is a shocking results. However, since we performed a qualitative study, our data is not suitable to assess how often this occurs in practice. Our study is a good basis to

investigate such issues in a quantitative manner. We have amended the discussion section (strengths and limitations) to address this point.

- I was confused by this statement:

“However, HCPs added the existence of peers was diverse and misinterpretation of patient to response about peer was commonly occurred.” Are you saying peer-to-peer support was diverse? Please explain what you mean by the entire statement.

Response: We have rephrased this statement to clarify the meaning, also in response to comments by reviewer 1.

We wrote: “HCPs said that the influence of peers may be positive or negative.” (Page 10 Line 30)

-Your statement, “Patients who had sufficient knowledge revealed that being healthy and being adherent reduced stigma from others” implies a threshold of knowledge. Did you assess level of knowledge? What does that mean in the context of your study? Same with “insufficient” level.

Response: We agree with your comments. We did not assess the level of knowledge of our participants. We revised our sentence and we deleted “sufficient knowledge” from our sentence.

We wrote: “Differently, the patient’s knowledge level played a role in coping with HIV.” (Page 8 Line 1).

-You use the word “Interestingly” to begin many sentences in the results section. Please use it judiciously.

Response: Thank for your comment. We revised the manuscript as you suggested and deleted that word from our manuscript.

-Discussion

Please explain what you mean that your findings are “more diverse than a previous study”. I’m confused by the word “diverse”.

Response: We have rephrased our sentences and we changed this paragraph.

We wrote: “In recent years, health care has been reorganized, so that HIV care can be provided in remote areas by Puskesmas, but some patients do not use those because of a lack of privacy. As in other studies, fear of disclosure of HIV status and the stigma associated with disclosure was an important barrier to accept this care closer to home [29]. This was not only found to be a barrier to get supply with ART, but also to discuss concerns and get information on the disease and treatment [28],[30].” (Page 15 Line 16)

-Your discussion would be stronger if you made clear and direct recommendations based on your findings to address the different levels of factors that influence ART adherence.

What organization changes need to be made?

How can health care providers improve?

How can PLHIV experience increased social support?

What strategies can improve the ART knowledge of PLHIV?

Response: We have amended the discussion section to include more recommendations.

-This is an important study. The policy and clinical implications should directly reflect your findings. Best of luck with your work.

Response: Thank you.

VERSION 2 – REVIEW

REVIEWER	Cynthia D. Fair Elon University USA
-----------------	---

REVIEW RETURNED	02-Jun-2020
-------------

GENERAL COMMENTS	Thank you for the opportunity to review your revised manuscript. I can tell you have improved the quality of your work. I have only a few remaining comments. Results Under “Verbal and nonverbal communication: I do not understand this sentence: “Participants stressed it was important to talk about genital terms in prudent ways...What does that mean? Was discussion of genitalia the only example of verbal communication that needed sensitivity? Further clarification would be helpful. I am still concerned that the following statement was not qualified. Did only one HCP report this? “It was commonly found that HCPs refused to treat patients and asked other colleagues to replace their shift in preparing medication.” I do not feel as though this has been adequately addressed in the discussion. Would mere education of HCPs address this violation of professional conduct? It seems to be beyond the training, but points to the necessity of enforcing policies. Best of luck with your important work.
---

VERSION 2 – AUTHOR RESPONSE

Reviewer: 2

Results

Under “Verbal and nonverbal communication:

I do not understand this sentence: “Participants stressed it was important to talk about genital terms in prudent ways...What does that mean? Was discussion of genitalia the only example of verbal communication that needed sensitivity? Further clarification would be helpful.

Response: In discussion part, we have already written that the culture and local value influenced Papuans whether to accept or to reject information particularly on HIV and sexuality. Participants highlighted there is a need to not mention genital and sex directly. It is common to use other words instead of genital term since genital is related to moral and pornography. Two references we used showed that demonstrate condom use and reproductive health program remains a tough problem to be accepted among Papuans.

We wrote: “Examples included HCPs laughing or not paying attention to patients or discussing sensitive issues like genitals directly and loudly, as shown in other studies [8], [34].

I am still concerned that the following statement was not qualified. Did only one HCP report this? “It was commonly found that HCPs refused to treat patients and asked other colleagues to replace their shift in preparing medication.”

Response: We found refusing to treat patients have been mentioned in our interviews with our participants however we merely highlighted one strong quote and use it to make our point easily accepted.

I do not feel as though this has been adequately addressed in the discussion. Would mere education

of HCPs address this violation of professional conduct? It seems to be beyond the training, but points to the necessity of enforcing policies.

Response: We amended that one of our inclusion criteria of HCP was having completed training in HIV care. We believed the professional conduct due to stigma seems absent in one of HIV training topics. According to your suggestion, we added one sentence in discussion part as a need to improve their professional while treating PLHIV.

We wrote: "Since having training before HCPs placed is mandatory, this study recommends stigma reduction should be available in one of training topics.